# High-Throughput Screening for Inhibitors of the SARS-CoV-2 Protease Using a FRET-Biosensor

**DOI:** 10.3390/molecules25204666

**Published:** 2020-10-13

**Authors:** Alistair S. Brown, David F. Ackerley, Mark J. Calcott

**Affiliations:** School of Biological Sciences, Victoria University of Wellington, Wellington 6012, New Zealand; alistair.brown@vuw.ac.nz (A.S.B.); david.ackerley@vuw.ac.nz (D.F.A.)

**Keywords:** SARS-CoV-2, COVID-19, 3CLPro, cysteine protease, ebselen, apomorphine, aporphine

## Abstract

The global SARS-CoV-2 pandemic started late 2019 and currently continues unabated. The lag-time for developing vaccines means it is of paramount importance to be able to quickly develop and repurpose therapeutic drugs. Protein-based biosensors allow screening to be performed using routine molecular laboratory equipment without a need for expensive chemical reagents. Here we present a biosensor for the 3-chymotrypsin-like cysteine protease from SARS-CoV-2, comprising a FRET-capable pair of fluorescent proteins held in proximity by a protease cleavable linker. We demonstrate the utility of this biosensor for inhibitor discovery by screening 1280 compounds from the Library of Pharmaceutically Active Compounds collection. The screening identified 65 inhibitors, with the 20 most active exhibiting sub-micromolar inhibition of 3CL^pro^ in follow-up EC_50_ assays. The top hits included several compounds not previously identified as 3CL^pro^ inhibitors, in particular five members of a family of aporphine alkaloids that offer promise as new antiviral drug leads.

## 1. Introduction

Severe acute respiratory syndrome coronavirus 2 (SARS-CoV-2), the causative agent of the current coronavirus disease 2019 (COVID-19) outbreak, was first identified in late 2019. SARS-CoV-2 infections result in a range of symptoms including: loss of smell and taste, persistent cough and chest pain [1]. As of September 2020, there have been over 31 million confirmed cases of COVID-19 and approximately 970,000 deaths [2]. The spread of COVID-19 currently shows no signs of slowing and while several promising vaccines are in clinical development [3], all are yet to complete clinical trials and begin distribution to the general population. It is therefore of paramount importance to rapidly develop and/or repurpose safe and effective drugs to treat SARS-CoV-2 and reduce the global burden of this pandemic.

SARS-CoV-2 is a member of the *Coronavirus* family and belongs to the *Betacoronavirus* genus, the same genus as other notable human pathogens including severe acute respiratory syndrome coronavirus (SARS-CoV, with which it shares approximately 80% sequence identity [4]) and Middle East respiratory syndrome coronavirus. SARS-CoV-2 is a positive-sense, single stranded RNA virus with a genome consisting of approximately 30,000 nucleotides [5]. Two thirds of the genome consists of two open reading frames called ORF1a and ORF1b, which are translated using a programmed ribosomal frameshift into two polyproteins—pp1a and a C-terminus extended form pp1ab [6]. The two polyproteins contain 16 non-structural proteins, which have critical roles in viral replication. The two proteases, 3-chymotrypsin-like cysteine protease (3CL^pro^), also known as the main protease, and papain-like protease are released auto-catalytically and cleave pp1a and pp1ab into the functional proteins [7] (Figure 1). 3CL^pro^ is a homodimer and is structurally highly similar to 3CL^pro^ from SARS-CoV [4,8] (Figure 1). 3CL^pro^ recognises a cleavage site of X-(L/F/M)-Q↓(G/A/S)-X, wherein X represents any amino acid and ↓ represents the cleavage site [9]. No known human proteases recognise the same cleavage site, offering prospects for the identification of inhibitors with low toxicity profiles [9].

Screening for inhibitors of SARS-CoV-2 3CL^pro^ has previously been conducted in high-throughput using a chemically-synthesised fluorophore and quencher separated by a cleavable peptide sequence [8]. Testing of 10,000 compounds with this probe identified seven hits, with ebselen being the strongest inhibitor. However, the specialised nature and cost of chemically synthesised probes makes them inaccessible to use in high-throughput screens for many facilities [10]. In contrast, protein-based biosensors can readily be prepared using equipment available in most molecular biology labs. A fluorescence resonance energy transfer (FRET)-based biosensor containing cyan-fluorescent protein (CFP) and yellow-fluorescent protein (YFP) as well as a luciferase-based biosensor have previously been used to determine the substrate specificity of 3CL^pro^ from SARS-CoV [11] and MERS-CoV [12], respectively. We reasoned that it should be possible to use a similar design to construct a biosensor capable of accurately and sensitively reporting on drug-mediated inhibition of SARS-CoV-2 3CL^pro^.

Here we report our development, optimisation and application of a high-throughput screen and EC_50_ assay using a protein-based FRET-biosensor to identify inhibitors of 3CL^pro^ from SARS-CoV-2. The biosensor is easy to express and purify from *Escherichia coli*, making it cheap and accessible for molecular biology laboratories, and the screen is robust, with a high level of correlation between replicates.

## 2. Results

### 2.1. Proteolysis of an eCFP-Venus Biosensor by SARS-CoV-2 3CL^pro^

To create a FRET-based biosensor, the fluorescent proteins eCFP and Venus were selected because they generate a FRET signal with a large dynamic range [13]. A T7 promoter driven, N-terminus His_6_ construct was made to express eCFP and Venus linked by the peptide sequence TSAVLQ↓SGFRK. This peptide linker contains the cleavage site found immediately upstream to 3CL^pro^ in its native polyprotein [8] (Appendix A). The purpose of the linker is to act as an on/off switch, holding the two fluorescent proteins in sufficiently close proximity for efficient FRET, with cleavage by 3CL^pro^ separating the proteins and thereby eliminating the signal at 528 nm (Figure 2A). The recombinant eCFP-Venus biosensor expressed well with no solubility or toxicity issues, and was purified using Ni-NTA chromatography from triplicate 400 mL *E. coli* BL21(DE3) cultures with a mean yield of 38.1 ± 1.4 mg of protein.

To assess whether treatment with protease reduced FRET in a quantifiable fashion, reactions containing 500 nM of the eCFP-Venus biosensor and 25 nM of SARS-CoV-2 3CL^pro^ were incubated at 30 °C. The ratio of the emission maxima of the acceptor (Venus) to donor (eCFP) (R_528/477_) was calculated from measurements taken at 30-min intervals (Figure 2B). This showed a high signal stability in the absence of protease and a time-dependent decrease following protease treatment. After 18 h incubation, samples were analysed using SDS-PAGE (Figure 2C). The replicates without protease produced a band corresponding to intact eCFP-Venus biosensor. Treatment with protease caused this band to almost completely disappear and a band corresponding to the similarly sized monomers of eCFP and Venus to appear. Consistent with emission from Venus diminishing as the linker between eCFP and Venus was cleaved, complete spectra recorded at 18 h showed that protease treatment had caused a large decrease at 528 nm, i.e., the emission maximum of Venus (Figure 2D). Collectively, these data indicated that the concentration of the eCFP-Venus biosensor used in these assays allowed robust measurement of protease activity and confirmed its suitability for high-throughput screening. Given our average yield of 38 mg eCFP-Venus biosensor per 400 mL culture, this concentration allows for approximately 6800 reactions to be performed from a single culture.

### 2.2. Characterisation and Application of the eCFP-Venus Biosensor for High Throughput Screening

As a proof of concept, the eCFP-Venus FRET-biosensor was used to screen the library of pharmacologically active compounds (LOPAC^®1280^, Sigma-Aldrich, St. Louis, MO, USA) for 3CL^pro^ inhibitors, on the presumption that an inhibitor would block cleavage and result in high R_528/477_ values, similar to the no protease controls from Figure 2B–D. An endpoint of 4 h was selected for measuring inhibition (Figure 2B), seeking to maximise the difference between digested and undigested eCFP-Venus biosensor while allowing the detection of weaker inhibitors by not letting the reaction run to completion. The LOPAC^®1280^ compounds were screened in duplicate at 5 µM in 96-well plates. Each plate also contained eight no-protease controls (representing 100% protease inhibition) and eight no-inhibitor controls representing 0% inhibition (Appendix A).

We first assessed the quality of the data using the metric Z’, which is based on the separation of positive and negative controls and is a key measure of quality for high-throughput screens [14]. A Z′ value above 0.5 is considered an excellent result and represents 12 standard deviations of separation between the controls. Here, the relative percentage inhibition for the no protease (M = 100.00, SD = 2.07) and protease controls (M = 0.00, SD = 1.14) from each plate revealed a clear separation band with a Z’ of 0.9 (Figure 3). This confirmed the signal distinction was more than sufficient for high-throughput screening.

We next examined the consistency between duplicate wells for each of the LOPAC^®1280^ compounds, observing a high level of reproducibility with an R^2^ value of 0.97 (Figure 4A; Appendix A). The mean level of inhibition was close to 0% for most compounds. Setting a threshold of 19% inhibition, equivalent to a Z’ score of 0.5, identified 65 compounds that were inhibiting 3CL^pro^ (Figure 4B). The library was then counter-screened without protease to assess the frequency at which compounds might interfere with FRET (Figure 4C). None of the 65 hits were identified as inhibiting FRET and, of the 1,280 compounds screened, only three (NF 023, anthrapyrazolone and dipyridamole) caused a large reduction in fluorescence. The high R^2^ value between replicates shows the assay to be robust at identifying hits, and screening in the absence of protease confirms that the false positive rate for compounds interfering with FRET is low and can be rapidly eliminated by counter screening in the absence of protease.

The 65 compounds above the 19% inhibition threshold were tested further in dose–response assays. All compounds exhibited a dose–response and the top 22 were found to have an EC_50_ of below 1 µM (Appendix A). Whereas the primary screen contained Triton X-100 at 0.01%, the dose–response assays were performed both with and without Triton X-100. A decrease in EC_50_ due to Triton X-100 is indicative of false positives due to compounds forming aggregates and sequestering proteins [15]. Two of the top hits, Disulfiram and SCH-202676, exhibited >2-fold reduction in levels of inhibition upon addition of 0.01% Triton X-100—an observation consistent with these being aggregation-based false positives [15]. The remaining 20 compounds each gave EC_50_ values within the range 27 nM to 990 nM (Figure 5; Appendix A).

## 3. Discussion

### Evaluation of Inhibitors Detected in This Screen

The top hit in our screen with an EC_50_ of 27 nM was ebselen, which has previously been identified as an effective inhibitor of SARS-CoV-2 3CL^pro^ [8]. Ebselen reacts with thiol residues to form a selenosulfide bond and a general reactivity with thiols could explain its inhibitory activity against the 3CL^pro^ cysteine protease [17]. Supporting this presumption, another generically thiol reactive compound (4-chloromercuribenzoic acid) was the third most effective inhibitor in our assays, with an EC_50_ of 85 nM. Although ebselen was able to prevent viral replication in vitro [8], it has been suggested that its general thiol reactivity might preclude inhibition of 3CL^pro^ in vivo, as the selenosulfide linkage might be reduced by cellular reactants [17]. However, ebselen appears to only covalently modify a portion of 3CL^pro^ [8], and molecular simulations have identified a second putative binding pocket in a region essential for dimerisation of 3CL^pro^. Thus, its inhibitory mechanism may not be reliant on selenosulfide bond formation [18]. While further work remains to confirm whether ebselen is a plausible therapeutic treatment, our identification of ebselen as the top hit in this study both validates our screen and supports that it is a potent inhibitor of 3CL^pro^.

The second strongest inhibitor identified in our screening, with an EC_50_ of 81 nM, was PD 404,182. This is an antibiotic against Gram-negative bacteria that is known to inhibit 3-deoxy-*d*-manno-octulosonic acid 8-phosphate synthase [19]. PD 404,182 has also previously been shown to be a highly potent inhibitor of HCV and HIV by causing physical disruption of the virion [20,21], and its ease of synthesis suggests it is tractable for use as an antiviral [22]. It has additionally been identified as an irreversible inhibitor of dimethylarginine dimethylaminohydrolase 1 and histone deacetylase 8 via irreversible binding to cysteine residues [23,24]. This may suggest promiscuous binding to the cysteine residue of 3CL^pro^ as a mechanism of inhibition. Nevertheless, the nanomolar activity towards 3CL^pro^ is particularly potent. As PD 404,182 also has potential to disrupt virus particles, this compound might target multiple stages of the SARS-CoV-2 life cycle.

Another potentially explicable hit identified in our screen was *Z*-l-Phe chloromethyl ketone (ZPCK). Peptidyl chloromethyl ketones are known potent inhibitors of cysteine proteases via formation of an irreversible thioether adduct, with target specificity conferred by the peptide sequence [25]. Although mainly used for biochemical investigations, a similar strategy of linking the core structure of a peptide or peptidomimetic to a reactive warhead has successfully been used for treating hepatitis C [26] and multiple myeloma [27]. Here, ZPCK exhibited an EC_50_ of 230 nM, substantially lower than the analogues Tosyl-l-Phe chloromethyl ketone (TPCK; EC_50_ of 1.74 µM) and Tosyl-l-Lys chloromethyl ketone (TLCK; EC_50_ of 10.63 µM) (Appendix A). These compounds differ from ZPCK in substituting the carboxybenzyl to a tosyl group, with an additional substitution of Phe to Lys for TLCK, both substitutions appearing to be detrimental to 3CL^pro^ inhibition. The ability to modulate the level of 3CL^pro^ inhibition by varying the peptide sequence suggests that further structure-activity-relationship studies could improve selectivity of these, and other, peptide-warhead compounds.

Five of the top twenty hits identified in our screening were apomorphine or analogues thereof (*N*-allylnorapomorphine, *R*(−)-propylnorapomorphine, *R*(−)-2-hydroxyapomorphine, and *R*(−)-2,10,11-trihydroxy-*N*-propylnoraporphine). Apomorphine is an aporphine alkaloid used as a treatment for Parkinson’s disease that is being explored for various additional indications including other neurological disorders, erectile dysfunction and cancer [28]. The compounds may be particularly relevant for lung infections as apomorphine can be safely administered via inhalation [29]. The five analogues identified here differ in *N*-alkylation as well as hydroxylation at the 2-position. The compounds were tolerant to both these modifications with EC_50_ differing at most by two-fold (346 nM to 746 nM). In contrast, the 10-hydroxyl group appears to be essential for activity. *R*(−)-apocodeine, which only differs from apomorphine by methylation at this position, exhibited only negligible (6%) inhibition of 3CL^pro^ in the 5 µM primary screen, failing to reach the threshold for testing in dose–response assays.

Finally, whereas the plant flavonoid myricetin was found to have an EC_50_ of 820 nM, the related compounds quercetin, dihydroquercetin and luteolin caused only 1.9%, 7.5% and 1.3% inhibition, respectively, in the initial screen. Each of these flavonoids comprise a bicyclic chromone linked to a phenyl ring (the ‘B-ring’), which contains three hydroxyl groups in the case of myricetin and two for quercetin, dihydroquercetin and luteolin. The additional hydroxyl is clearly important, as it is the only difference between quercetin and myricetin. Although not detected as an inhibitor at the concentrations we tested, both quercetin and its *O*-glycoside quercetin-3-β-galactoside have previously been identified as weak inhibitors of SARS-CoV 3CL^pro^, with IC_50_ values of 23.8 μM and 42.8 μM, respectively [30,31]. In silico docking found the 3′ hydroxyl group of quercetin-3-β-galactoside forms a hydrogen bond with Q189 of SARS-CoV 3CL^pro^ [31], and removal of this hydroxyl group substantially reduces inhibitory activity of both quercetin and quercetin-3-β-galactoside [30,31]. Being connected by a rotatable bond, the symmetry of the B ring means either the 3′ or 5′ hydroxyl groups of myricetin has the potential to interact with Q189. This flexibility may explain the increased potency over quercetin, with the additional hydroxyl group increasing the likelihood of myricetin being in a conformation to form a hydrogen bond with Q189.

## 4. Materials and Methods

### 4.1. Protein Expression

Genes encoding the eCFP-Venus biosensor (Appendix A) and SARS-CoV-2 3CL^pro^ (Appendix A) were codon-optimized and cloned into the *Nco*I and *Xho*I sites of plasmid pET28a(+) by TWIST Bioscience (Illumina; San Diego, CA, USA). The N-terminus of 3CL^pro^ contained the cleavage site natively found between non-structural protein 4 and 3CL^pro^ (TSAVLQ↓SG) to allow a natural terminus to form by autocleavage, and the C-terminus contained two added residues (GP) and a His_6_-tag. The plasmids were used to transform *E. coli* BL21(DE3) and subsequently purified by miniprep and sequence verified by Macrogen, Inc (Seoul, South Korea).

To purify proteins, a 400 mL culture of *E. coli* was grown in lysogeny broth at 37 °C, 200 rpm until an OD_600_ of 1.6 was reached. The culture was then incubated on ice for 20 min. Isopropyl-d-thiogalactoside (IPTG) was added at a concentration of 1 mM. The cultures were then incubated at 18 °C, 200 rpm for 18 h for the eCFP-Venus biosensor and 37 °C, 200 rpm for 5 h for 3CL^pro^. Cells were harvested by centrifugation and the pellet frozen. The pellet was subsequently thawed and resuspended in 30 mL of buffer A (20 mM Tris, 500 mM NaCl, 5 mM imidazole, pH 8.0). For SARS-CoV-2 3CL^pro^ the cells were lysed by sonication on ice, and for the eCFP-Venus biosensor the cells were passed twice through a French press cell disruptor (Thermo Electron; Franklin, MA, USA) at 1000 psi. The insoluble material was removed by centrifugation at 18,000× *g* for 30 min. The supernatant was loaded onto a column containing 2 mL of Ni-NTA His-Bind Resin (Novagen; Darmstadt, Germany). The column was washed with 10 mL of buffer A, followed by 40 mL of buffer A for the eCFP-Venus biosensor, or 20 mL of buffer B (20 mM Tris, 500 mM NaCl, 60 mM imidazole, pH 8.0) for SARS-CoV-2 3CL^pro^. Bound protein was eluted using 8 mL of buffer C (20 mM Tris, 500 mM NaCl, 1 M imidazole, pH 8.0). The flow through was subjected to buffer exchange using buffer D (50 mM Tris, 1 mM EDTA, pH 7.5) using an Amicon Ultra 15 centrifugal filters (30 kDa cutoff, Merck Millipore) and glycerol added to a final concentration of 40% (*v*/*v*). Absorbance was measured at 280 nm using a NanoPhotometer^®^ NP80 (Inplen; Munich, Germany) and used to determine the concentration based on an extinction coefficient of 49,530 M^−1^ cm^−1^ and 33,640 M^−1^ cm^−1^ calculated using the ExPASy ProtParam tool [32] for the eCFP-Venus biosensor and SARS-CoV-2 3CL^pro^, respectively.

### 4.2. Enzymatic Activity and Inhibition Assays

All assays of protease activity were performed in 96-well plate format with a total volume of 200 µL of 50 mM Tris, 1 mM EDTA, pH 7.5. Each well contained 25 nM of SARS-CoV-2 3CL^pro^ and 500 nM eCFP-Venus biosensor. Assays were set up by first adding 3CL^pro^ plus compound in a 100 µL volume of buffer to each well, and then initiated by the addition of the eCFP-Venus biosensor in a 100 µL volume. The plates were incubated at 30 °C for 4 h. Fluorescence was measured using an excitation wavelength of 434 nm and emission wavelengths of 477 nm and 528 nm. High-throughput screening was performed in duplicate using 5 µM of each compound. Z’ score was calculated using the sample means and standard deviations for the positive and negative controls. The dose–response for selected hits was performed using a concentration ranging from 0.001 µM to 20 µM, and included a sample with no added compound. All data processing was performed in R, version 4.0.2 and scripts are available at: https://github.com/MarkCalcott/Protease_screen

### 4.3. Chemicals

The LOPAC^®^1280 (International Edition) was used for screening and was purchased from Sigma-Aldrich. All compounds were supplied as stock solutions at 10 mM and diluted in dimethyl sulfoxide to 500 µM for primary screening or 1 mM for dose-response assays.

## 5. Conclusions

This work has validated our eCFP-Venus biosensor as being robust and amenable to high-throughput screening and identified several new inhibitors of SARS-CoV-2 3CL^pro^. Expression levels of the biosensor were high, and each purification gave sufficient yield to screen thousands of compounds. The assay met the benchmarks for a reliable high-throughput screen, with a high Z’ score, low levels of variation between replicates, and relatively few compounds interfering with the FRET signal. That the top inhibitor in this study was ebselen demonstrates the consistency of our assay with previous screening using chemically synthesised probes, as well as validating ebselen as a particularly potent inhibitor. Our screening has also identified 19 other compounds that have EC_50_ values below 1 µM, five of which were aporphine alkaloids not previously identified as 3CL^pro^ inhibitors, suggesting a promising starting point for structure-activity-relationship studies to develop new antiviral compounds. A key advantage of this screen over chemically synthesised probes is that it uses the same equipment that would be required for any laboratory wanting to purify and test 3CL^pro^, making high-throughput screening easily accessible for other researchers.

## Figures and Tables

**Figure 1 molecules-25-04666-f001:**
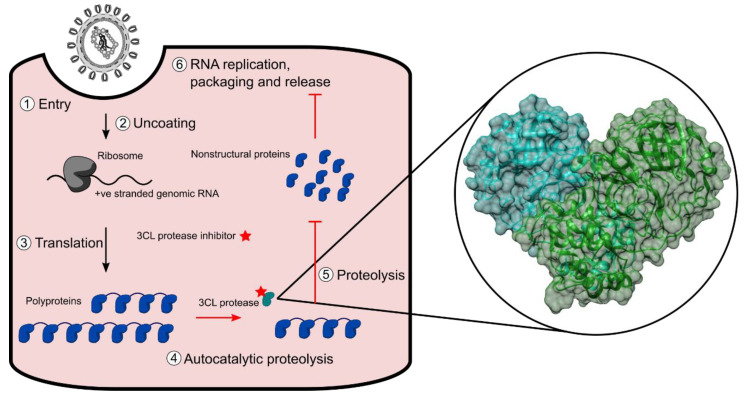
Schematic representation of the SARS-CoV-2 lifecycle in a host cell and the interruption by a 3CL^pro^ inhibitor. The insert is the crystal structure of 3CL^pro^ derived from PDB: 6M2Q.

**Figure 2 molecules-25-04666-f002:**
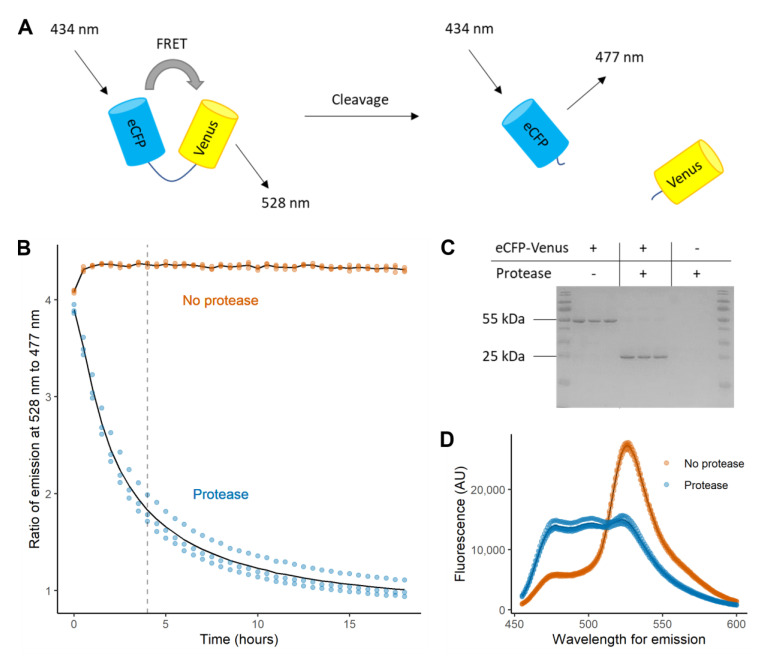
(**A**) When eCFP is excited at 434 nm in the intact biosensor, the close proximity of eCFP and Venus results in FRET emission from Venus at 528 nm. Cleavage of the linker by SARS-CoV-2 3CL^pro^ stops FRET occurring and emission now comes directly from eCFP at 477 nm. (**B**) The dynamic ratio of emission from the FRET acceptor (Venus) and donor (eCFP)(R_528/477_) during treatment with (blue) or without (orange) SARS-CoV-2 3CL^pro^. The emission of the FRET donor (λ_ex eCFP_ 477 nm) and acceptor (λ_ex Venus_ 528 nm) after excitation at 434 nm were measured at 30 min intervals and used to calculate the R_528/477_. A dashed line at 4 h indicates the time used in endpoint assays for subsequent high-throughput screening. (**C**) A 12% SDS-PAGE gel of samples taken from the reactions measured in Panel B at 18 h. Each well in the SDS-PAGE gel contained a 12 µL sample taken from a 200 µL reaction. The presence of the eCFP-Venus biosensor and protease is indicated above the gel. Bands present at 55 kDa are consistent with the eCFP-Venus biosensor, including intact linker and His_6_-tag (predicted mass of 56 kDa), and bands just above 25 kDa are consistent with monomers of similarly sized eCFP (predicted mass of 27.5 kDa) and Venus (28.5 kDa). (**D**) Emission spectra of the eCFP-Venus biosensor after 18 h of treatment with (blue) and without (orange) protease. Experiments were performed in triplicate using separately purified eCFP-Venus biosensor and SARS-CoV-2 3CL^pro^. The average between replicates is represented by a line and all datapoints are shown as dots.

**Figure 3 molecules-25-04666-f003:**
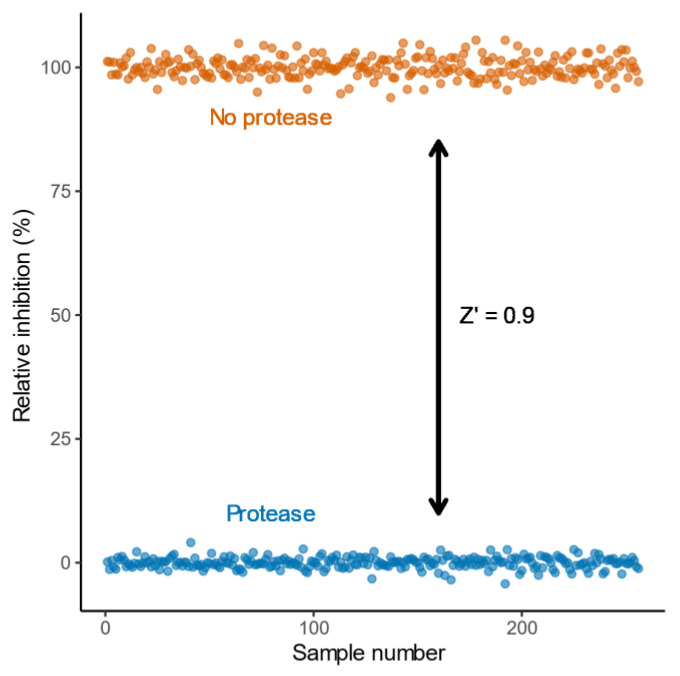
Separation of the positive and negative controls used to calculate relative inhibition of protease. Relative inhibition was calculated using the average R_528/477_ for the protease treated controls as 0% inhibition and the average R_528/477_ for untreated controls as 100% inhibition. The data are the 256 no protease controls and 256 protease controls collected from all plates when screening the LOPAC^®1280^ in duplicate. All datapoints are shown.

**Figure 4 molecules-25-04666-f004:**
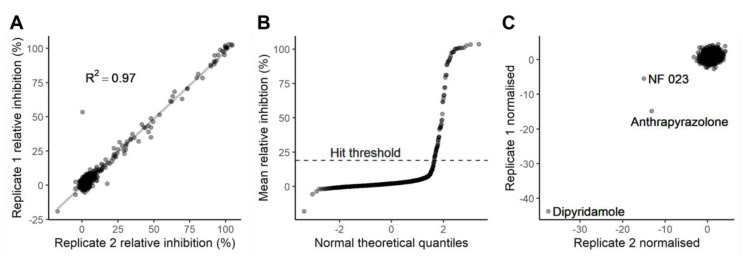
Screening the LOPAC^®1280^ in duplicate. (**A**) Consistency between duplicates during screening. Each axis represents one replicate. The relative inhibition of protease activity for compounds was calculated as per Figure 3. (**B**) A quantile-quantile plot showing mean relative inhibition versus normal theoretical quantiles. The 19% relative inhibition used as a threshold for hit selection is indicated with a dashed line. (**C**) The effect of compounds on the R_528/477_ in the absence of protease. Each axis represents one replicate and R_528/477_ was normalised for each plate using sixteen eCFP-Venus biosensor (no protease) controls. Duplicate experiments were performed using separately purified eCFP-Venus biosensor and SARS-CoV-2 3CL^pro^, and all datapoints are shown.

**Figure 5 molecules-25-04666-f005:**
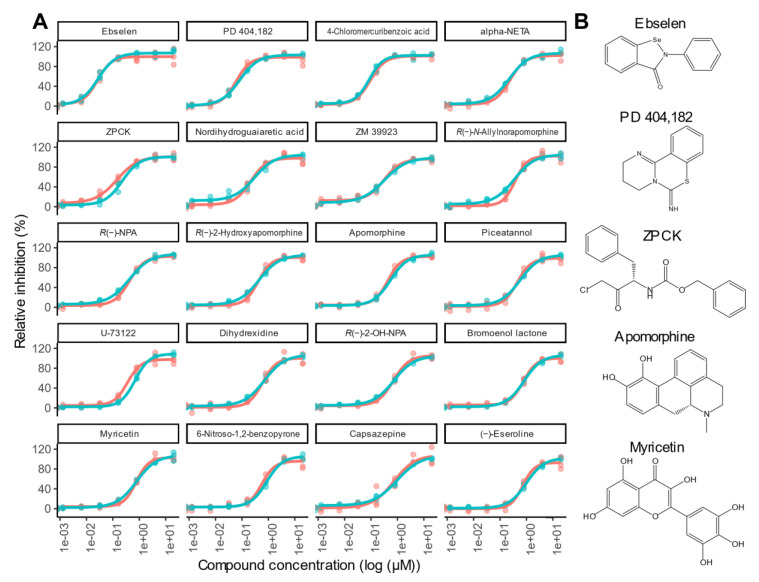
Dose–response curves for the top 20 inhibitors identified in this study and selected structures for compounds discussed in the text. (**A**) Dose–response assays were performed both with (blue) and without (red) 0.01% Triton X-100. Each compound was assessed across a 5-fold serial dilution series starting at 20 µM. Relative inhibition was calculated using the average R_528/477_ for the protease treated controls as 0% inhibition and the average R_528/477_ for the no protease controls as 100% inhibition. Experiments were performed in triplicate using a different preparation of purified eCFP-Venus biosensor for each repeat. EC_50_ values were calculated using the R package ‘dra’ [16], and all data points are shown. (**B**) Structures of five compounds that are discussed further in the text. Compounds tested in this Panel A are: Ebselen; PD 404,182; 4-Chloromercuribenzoic acid; 2-(alpha-Naphthoyl)ethyltrimethylammonium (alpha-NETA); *Z*-l-Phe chloromethyl ketone (ZPCK); Nordihydroguaiaretic acid; ZM 39923; *R*(−)-*N*-Allylnorapomorphine; *R*(−)-Propylnorapomorphine (*R*(−)-NPA); *R*(−)-2-Hydroxyapomorphine; Apomorphine; Piceatannol; U-73122; Dihydrexidine; *R*(−)-2,10,11-Trihydroxy-*N*-propylnoraporphine (*R*(−)-2-OH-NPA); Bromoenol lactone; Myricetin; 6-Nitroso-1,2-benzopyrone; Capsazepine; (−)-Eseroline.

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
