# Peer review of "High-Throughput Screening for Inhibitors of the SARS-CoV-2 Protease Using a FRET-Biosensor"

_molecules, 2020, doi:10.3390/molecules25204666_

Round 1

Reviewer 1 Report

I would suggest releasing it with minor technical fixes and minor revisions:

Line 39: use a comma between “proteins” and “which”

Line 44: “X[LFM]Q↓[GAS]X” in the paper [9] cited was “X-(L/F/M)-Q↓(G/A/S)-X”

Line 65: “expressed and purifable” better write “easy to express and purify”

Line 73: “TSAVLQ|SGFRK” better write “TSAVLQ↓SGFRK” to be consistent with line 44

Line 76: please specify “the signal at 528 nm”

Line 93: “mass of 56.0 kDa” better write “56 kDa”

Line 95: “(D) Emission spectra” please show the pure emission spectra in Figure 2D instead of relative spectra

Line 101: “and 25 nm” must be “and 25 nM”

Line 121: “Figures 1B-1D” must be “Figures 2B-2D”

Line 125: “containing” better write “contained”

Line 176: “select” better write “selected”

Reviewer 2 Report

This work reports the development, optimization and application of a high-throughput screen and EC50 assay using a protein-based FRET-biosensor to identify inhibitors of 3CLpro from SARS-CoV-2. They claimed the proposed biosensor is readily expressed and purifiable from E. coli, making it cheap and accessible for molecular biology laboratories. The screen is robust and with a high level of correlation between replicates. The article structures well and the content is complete. However, wrong spellings appear throughout the article, please check and revise. Several sentences are too long and the punctuations are wrong, please check and revise. Some examples are list as below.

Line 43,45  wrong words spelling “recognises” should recognizes.

Line 52 “chemically-synthesised” should be chemically-synthesized.

Line 54 “specialised nature and cost of chemically synthesised probes” should be specialized nature and cost of chemically synthesized probes.

Line 82 what is the meaning of the word “promity”?

Line 84 “occuring” should occurring
